# Polysaccharides as Support for Microbial Biomass-Based Adsorbents with Applications in Removal of Heavy Metals and Dyes

**DOI:** 10.3390/polym13172893

**Published:** 2021-08-27

**Authors:** Alexandra Cristina Blaga, Carmen Zaharia, Daniela Suteu

**Affiliations:** 1Department of Organic, Biochemical and Food Engineering, ‘Cristofor Simionescu’ Faculty of Chemical Engineering and Environment Protection, “Gheorghe Asachi” Technical University of Iasi, 73 D. Mangeron Blvd, 700050 Iasi, Romania; acblaga@tuiasi.ro; 2Department of Environmental Engineering and Management, ‘Cristofor Simionescu’ Faculty of Chemical Engineering and Environment Protection, “Gheorghe Asachi” Technical University of Iasi, 73 D. Mangeron Blvd, 700050 Iasi, Romania; czah@tuiasi.ro

**Keywords:** aqueous solutions, biosorbents, microbial biomass, polymeric supports

## Abstract

The use of biosorbents for the decontamination of industrial effluent (e.g., wastewater treatment) by retaining non-biodegradable pollutants (antibiotics, dyes, and heavy metals) has been investigated in order to develop inexpensive and effective techniques. The exacerbated water pollution crisis is a huge threat to the global economy, especially in association with the rapid development of industry; thus, the sustainable reuse of different treated water resources has become a worldwide necessity. This review investigates the use of different natural (living and non-living) microbial biomass types containing polysaccharides, proteins, and lipids (natural polymers) as biosorbents in free and immobilized forms. Microbial biomass immobilization performed by using polymeric support (i.e., polysaccharides) would ensure the production of efficient biosorbents, with good mechanical resistance and easy separation ability, utilized in different effluents’ depollution. Biomass-based biosorbents, due to their outstanding biosorption abilities and good efficiency for effluent treatment (concentrated or diluted solutions of residuals/contaminants), need to be used in industrial environmental applications, to improve environmental sustainability of the economic activities. This review presents the most recent advances related the main polymers such as polysaccharides and microbial cells used for biosorbents production; a detailed analysis of the biosorption capability of algal, bacterial and fungal biomass; as well as a series of specific applications for retaining metal ions and organic dyes. Even if biosorption offers many advantages, the complexity of operation increased by the presence of multiple pollutants in real wastewater combined with insufficient knowledge on desorption and regeneration capacity of biosorbents (mostly used in laboratory scale) requires more large-scale biosorption experiments in order to adequately choose a type of biomass but also a polymeric support for an efficient treatment process.

## 1. Introduction

The exacerbated water pollution crisis is a huge threat to the global economy, especially in association with the rapid development of industry; thus, the sustainable reuse of different treated water resources has become a worldwide necessity [1]. Lack of sufficient clean water is a growing challenge in many countries, due to industrial activities (including chemical, textile, pulp and paper, metallurgic, leather, polymeric, painting and coatings, food, and pharmacological activities) that generate pollutants into the receiving water resources, threatening the ecosystem. In this context, international organizations have reviewed and developed highly restrictive rules that assess the impact of chemical species, taking into account their effects on the environment and the quality of human life. Such rules set maximum permissible quantitative limits to avoid unpleasant or even dangerous effects. Thus, the prevention, development, or optimization of depollution techniques/technologies, and materials for advanced wastewater treatment is both a priority and a continuous challenge for specialists in the field of environmental sciences [1,2,3,4,5,6,7,8].

Methods frequently applied in water depollution (precipitation, coagulation—flocculation, oxidation, reduction, ion exchange, membrane filtration, adsorption on charcoal or polymeric materials, electrochemical treatments, inverse osmosis, recuperation by evaporation, solvent extraction, etc.) are not efficient for total elimination of toxic metal ions or organic matter [9,10]. These methods are characterized by incomplete removal of metals and organic matter, limited tolerance to pH change, moderate or no selectivity for metals, requirement of high consumption of reagents, materials and energy, reduced efficiency at very high or low concentration levels of metals, and production of toxic sludge or other wasted products that also need treatment before disposal. To ensure a minimum negative impact of these issues, worldwide studies are focused on developing individual or combined methods of high efficiency and selectivity allowing simultaneous achievement of pre-concentration and analysis. Taking into consideration the trends of using natural materials for cleaning techniques of wastewaters, the specialists’ attention has been oriented to developing new techniques of bioremediation, as a part of the environmental biotechnologies. The advantages of these bioprocesses are that they use natural products, biodegradable, have low impact on the environment, and can remove a number of pollutants from various environmental matrices. The purpose of implementing such biotechnologies is to increase the efficiency of depollution processes based on the cost-benefit principle, with respect to the constant reduction of energy costs and manual labor, reducing the volume of raw materials handled, eliminating any sources of pollution, and lack of secondary production of hazardous waste [1,2,3,4,5,6,7,8,9,10].

Although it is a well-known process, (bio) adsorption consisting of the transfer of soluble substances from the aqueous medium on the surface of solid particles, with high porosity ((bio) adsorbent), is still a viable alternative due to its major advantages in terms of efficiency and cost. Biosorption using biomass (material of biological origin—microbial, vegetal, or animal cells containing in their structure different natural polymers such as polysaccharides and their derivates, proteins, lipids, etc.) as adsorptive materials, is a simple, useful, and effective treatment process [1,2,3,4,5,11,12,13,14,15].

The use of biological macromolecules presented in any biosystem (or organism) as support materials for effluent (e.g., wastewater) purification is a good approach due to the environmentally friendly properties and efficiency, but also to low cost, ease simplicity of design, or operation [16]. However, the large-scale use of these biological macromolecules is restricted, and numerous approaches have been studied for the development of cheap, effective, and ecofriendly biosorbents (materials that require as little processing as possible, and can be found in nature in large quantities, or can result as by-products from industry—wasted biomass from bioprocessing) capable of eliminating pollutants present in the industrial effluent (industrial wastewater) [15].

The main advantage of bioprocesses (biotechnological, microbiological, biological), compared to certain physico-chemical treatments, is given by the fact that over 70% of the organic matter expressed by COD_Cr_ can be converted into biosolids [6]. Thus, the depollution potential of different types of microorganisms opens directions to new emerging biotechnological processes to be further studied and transformed into practical and industrial solutions [1,2,3,4,5,6,7,8].

The purpose of this review is to present the types of polymers used to obtain biosorbents using different kinds of natural (living and non-living) or residual microbial biomass in immobilized or encapsulated forms.

## 2. Biosorption

During biosorption (Scheme 1), the pollutants from different effluents (wastewaters) are retained by binding/fixing them onto a material of biological origin (natural biomass-based biosorbent) [6,7]. The choice of the biomass used is extremely important: from agricultural byproducts or wood processing biomaterials (readily available and low cost since they have limited utility) to plants derived materials and microorganisms (usually metabolically inactive/non-living cells). All these biomaterials possess a chemical structure that allows sequestration of the pollutant, through different chemical bonds or physical interactions that can be formed especially at the cell wall level. Several pretreatment operations are required for the use of adsorbents based on biomass or their functionalization/activation: for plant derived wasted materials—chemical modifications using alkaline or acid solutions, or pyrolysis and microwave functionalization, while for microbial cells—chemical modification, carbonization, or grounding to increase the contact area [7,10,12,15,16,17,18]. An ideal biosorbent possesses the following characteristics: low economic values, high biosorption capacity (high affinity for pollutants), availability in large quantities, easy desorption of the retained pollutants, and multiple possible reuses [19,20,21].

The biomass (algae, bacteria, yeast, and fungi) can be used as living cells, offering the possibility of removing larger amounts of pollutants, since it has the enzymatic facility that allows it to either transform or degrade the pollutant (considered as food source for living cells/microorganism), but the biosorption process depends on the cell metabolism and requires specific conditions compatible with living cells (nutrient supply, temperature and pH in a certain domain, oxygen supply for aerobic cells, etc.) and can be stored for very limited periods of time [20,22,23]. The non-living biomass is used in the simpler and less expensive treatment processes, thus the biosorbent can be utilized for more than one cycle (taking into account the pollutant concentration), and no cell disruption occurs as long as the working conditions are not extreme. Several important advantages of using non-living/inactive cells are presented in Figure 1 [9,16].

The biosorption process is based on the characteristic of the microbial cell walls which consists of different polysaccharides, proteins, and lipids that provide a variety of functional groups (carboxylic, hydroxyl, phosphate, amino, thio etc.) that can interact with contaminants by diverse chemical forces [8,20].

The microbial biomass can be used in two ways for the biosorption process: free or immobilized. The differences between these two forms is presented in Table 1. Immobilization is commonly defined as a process of cell attachment and/or inclusion to or into a support (an inert coating that isolates and protects biomass cells from the external environment) [22].

There are different immobilization techniques that can be applied for biosorbents, with microbial cells being bonded either on the surface or within a polymer matrix: adsorption, covalent bonding, cross-linking, encapsulation, and entrapment in a matrix (Figure 2) [6,22,24,25,26].

**Table 1 polymers-13-02893-t001:** Advantages and disadvantages of free and immobilized biomass used as biosorbent [23,27].

Biomass	Free	Immobilized
Advantages	- high interfacial area; - high biosorption capacity (easily available functional groups) - no added cost related to immobilization	- easily regenerated; - higher mechanical stability; - higher resistance; - multiple uses; - incorporated into fixed and fluidized bed columns; - higher productivity; - easy separation of biomass and effluent; - avoidance of cell washouts.
Disadvantages	- small size and low density, - insufficient mechanical stability and low elasticity; - small size; - low density; - insufficient mechanical stability; - low elasticity; - difficult separation of phases (centrifugation and filtration); - difficult biosorbent regeneration.	- added cost related to immobilization; - higher mechanical diffusion resistance; - lower biosorbent capacity; - interaction between the carrier and the active sites of the biosorbent.

## 3. Polymer Support for Immobilization of Microbial Biomass to Obtain Biosorbents

An extremely important step in the immobilization process is the selection of a suitable support (it can be natural—gelatin, agar, gum, sodium alginate, calcium alginate, dextran, fats and fatty acids, starch, chitosan, sucrose, and wax, or synthetic—acrylic polymer and copolymers, and semi-synthetic—cellulose acetate, cellulose nitrate, ethylcellulose and hydroxypropylcellulose, methylcellulose, sodium carboxymethylcellulose, hydrogenated fat, and myristic alcohol), since it needs to fulfill several characteristics, presented in Table 2 [6,22,23,24,25].

The choice of a support is very important for the efficiency of immobilization and the stability of final biosorbent. The criteria for selecting a support are based on the physico-chemical properties of the support and biomass, but especially on the compatibility between these two components (the support should be insoluble and should not react with the biomass active groups involved in the biosorption), the processing and economic factors. Another criterion to consider is the desired size for the final particles. Through immobilization numerous particles are obtained, differentiated by their morphology and internal structure: (i) when the particle size is below 1 µm they are known as nanoparticles, nanocapsules, and nanosphere; (ii) the particles with a diameter between 3–800 µm are known as microparticles, microcapsules, or microspheres; (iii) particles larger than 1000 µm are called macroparticles [28], and (iv) particles larger than 10,000–100,000 um are named as ‘course’ particles. Their shape depends to a large extent on the structure of the immobilizing material and the obtaining/preparation method used.

Substances or materials used for immobilization must meet a number of specific conditions [29,30,31]:-To have adequate rheological properties (flow easily even at high concentrations).-To have the ability to disperse biomass.-Do not react with biomass either during the process or during storage.-To keep the biomass active throughout the immobilization process and then during the storage period.-To be able to be purified from solvents or other materials used during the immobilization process.-To maintain the maximum sorption capacity of the biomass.-To be insoluble in the effluent (wastewater).-Do not damage the cells.-To have low cost.

Choosing a support according to the desired application is an important task, which is why it must be considered significant the following conditions: toxicity, immobilization efficiency, stability, and microscopic properties of the particle surface, but also flexibility in overall shape, high diffusivity, simple immobilization procedure, and high biomass retention. Due to the impossibility of finding a single material that meets all these conditions, in practice, combinations of various polymers, or modifiers of support polymers properties are most often used [31,32,33].

The most important natural polymers and derivates that can be used for microbial immobilization are polysaccharides ([C_n_(H_2_O)_n_]_m_, where n = 6…8 and m = 40…30,000): cellulose, chitin, chitosan (deacetylated chitin) and alginate, presented in Table 3. These polysaccharides are characterized by biocompatibility, biodegradability, and non-toxicity, but some considerations related to mechanical straight, stability, and standardized pore size or immobilization surface need to be considered [32,33]. Immobilization of microorganisms in naturally polymers (such as polysaccharides) can increase the biosorption capacity of the matrices (alginate, pectate, and synthetic cross-linked polymer) up to 12-fold when compared to the use of polymers alone [34]. The ability to adsorb heavy metals of immobilized microbial cells (*Pseudomonas putida*) in various matrices: alginate–PVA–CaCO_3_ (adsorption without cells—60% Pb(II) and 20% Cd(II)) and carboxymethylcellulose (CMC) (adsorption without cells—3% Pb(II) and 5% Cd(II)) was compared and an increase of metal removal efficiency in all matrices after bacterial immobilization was observed: 75.5% Pb(II)/75% Cd(II) for alginate–PVA–CaCO_3_, and 32% Pb(II)/15% Cd(II) for cellulose support [35]. 

For polysaccharides the following issues must be taken into account in the choice of the support: monosaccharide composition and arrangement, position of glycoside linkages, rheological properties, and solubility. Bacterial cellulose (produced by acetic acid bacteria) is a multifunctional nano-biomaterial that offers high mechanical strength and purity, low cytotoxicity, and good biodegradability combined with a very high surface area (that improves their adhesion properties, useful in adsorption as immobilization technique). It has been used as an appropriate immobilization carrier/polymeric support for different microorganism: yeasts—*Saccharomyces cerevisiae*, *Yarrowialipolytica* [41], bacteria—*Lactobacillus* spp. [42], *Pseudomonas stutzeri* [43], and *Corynebacterium glutamicum* [44], but also as an individual biosorbent for fluoride [45] or other metals. Recently, *Komagataeibacterxylinus X-2* was immobilized in bacterial cellulose and proved improved mechanical properties (the bacterial cells act as consolidation points which connect numerous cellulose nanofibers) compared to simple bacterial cellulose films and higher adsorption capacities for Pb(II), Cu(II), Ni(II), and Cr(VI) due to the presence of amide groups in bacteria [46]. *Pseudomonas stutzeri* was immobilized in bacterial cellulose and used for nitrate removal from industrial effluent (wastewater) and contaminated groundwater and proved increased adsorption capacity, decreased cell leakage from the beads, and higher activity of immobilized cells [43].

For increasing *cellulose* adsorption efficiency, but also its surface area, roughness of surface morphology, thermal stability, and mechanical strength, chemical modification such as carboxymethylation, can be used. *Trametes versicolor*, both active and inactive (heat-treated), was used for the removal of Cu(II), Pb(II), and Zn (II) from aqueous solutions in immobilized forms using CMC beads. The biosorption capacity were found to be for Cu(II)—1.51 (active) and 1.84 mmol (inactive), Pb(II)—0.85 (active) and 1.11 mmol (inactive), and Zn(II)—1.33 (active) and 1.67 mmol (inactive), higher values being obtained for heat-inactivated biomass for each cation [47]. *Pannonibacterphragmitetus LSSE-09* was used in encapsulated form as liquid-core alginate–carboxymethyl cellulose capsules (optimum conditions for encapsulation: 30-min gelation time, 0.5% *w*/*v* sodium alginate, 2% *w*/*v* sodium carboxymethyl cellulose, and 0.1 M CaCl_2_) for the reduction of Cr(VI) to Cr(III) under alkaline conditions obtaining 4.2 mg/g.min reduction rate [48]. *Phanerochaetechrysosporium* (a white-rot fungus) and *Lentinussajor-caju* (fungus) were separately immobilized in CMC beads, stable under experimental conditions and used for the biosorption of Hg(II) and, respectively, Cr(VI) ions, being able to retain 100 mg/g of beads, 32.2 mg/g (d.w.) biomass, respectively [49,50]. Basidiospores of *Phanerochaetechrysosporium* (active and heat-inactivated immobilized on CMC) were used by different authors [51] for the removal of Hg(II) ions using aqueous solutions in the concentration range of 30–700 mg/L, the biosorption efficiency increasing with pollutant concentration increase. The experimental obtained results showed an increased biosorption capacity of the heat-treated biomass (inactive) compared to the active one, and especially compared to the polymeric support (CMC): 102.15 mg/g, 183.10 mg/g and 39.42 mg/g Hg(II), respectively [51]. *Aspergillus fumigatus* immobilized in sodium carboxymethylcellulose beads were used to remove reactive Brilliant Red K-2BP from aqueous solutions, obtaining the following optimum conditions: initial pH = 6–9, temperature = 40 °C, agitation rate = 150 r/min, beads diameter = 2.0 mm, and beads dosage = 3.0% [52].

***Alginates*** are natural polysaccharides from seaweed consisting of linear copolymers of two units: β-(1–4)-D-manuronic acid and β-(1–4)-L-guluronic acid (acids extracted from brown algae). Based on sodium alginate/natural or synthetic polymers, several types of biosorbents can be obtained: hydrogels, microspheres, blends, microparticles, nano-gels, electrospun fibers, films, nanoparticles, and nanocomposites [53]. Their characteristics include being nontoxic, legally safe for human use, easy to handle, available in large quantities and inexpensive, but during the immobilization, the immobilized cells do not undergo changes in physicochemical composition and the gel characteristics (transparency and permeability) increase to a very large extent, recommending the use of alginates as support polymer for immobilization. One disadvantage must however be analyzed: chelating agents such as EDTA, citrate, lactate, phosphate, or antigelling cations (Na(I) and Mg(II)) dissolve gel [54,55,56]. It has been used to immobilize bacteria (*Pseudomonas koreensis* [57], *Bacillus subtilis* [58,59,60]), and fungi (*Trichoderma viride* [61], *Trichoderma asperellum* [62], *Aspergillus niger* [63,64]). *Bacillus subtilis* immobilized in calcium alginate was used in batch mode (contact time 3 h) for biosorption of Cd(II) ions, obtaining the biosorption capacity of 251.91 mg/g (for initial Cd(II) concentration of 496.23 mg/L) and optimum conditions: pH 5.92, temperature 45 °C, and 1 g/L biosorbent dose. The Cd(II) ions were desorbed using a 0.1 M solution of HCl and the system was stable for five cycles [65]. Residual biomass of *Bacillus* sp. immobilized in sodium alginate was used in batch and dynamic working regime for biosorption of Brilliant Red HE-3B reactive dye and was obtained the biosorption capacity of 588.235 mg/g at 20 °C in batch conditions [59]. *Saccharomyces cerevisiae* waste biomass immobilized in the form of gel beads in Ca-alginate, Ca-alginate with graphene oxide, and polyvinyl alcohol-Ca-alginate-graphene oxide in CaCl_2_-boric acid solution was used for U(VI) removal from aqueous solutions. The best results (biosorption capacity of 21–35 mg U/g and the recovery with 0.1 M HNO_3_ solution of U(VI) ions was 91%) were obtained with beads obtained from 5% PVA-1% SA-2% yeast-0.01% GO-2% CaCl_2_-saturated boric acid [66]. These results were much lower than the ones obtained for suspended *Saccharomyces cerevisiae* biomass—biosorption capacity of 127.7 mg U/g dry biomass under pH 4.5 and initial concentration 10 to 1000 mg U/L [67]. For support polymer beads the maximum capacity of U(VI) biosorption was 23.4–31.4 mg/g, for 0–400 mg U/L initial concentration and 3.0 initial solution pH and 40 mg sorbent/g support [68]. Additionally, the biomass of *Saccharomyces cerevisiae* immobilized in sodium alginate and present in the form of gel beads was also used for the retention of organic dyes (Brilliant Red HE-3B, Methylene Blue, Orange 16, Rhodamine B), pollutants in the effluents from the textile industry [69]. Biosorption studies using immobilized yeast to retain the Brilliant Red HE-3B dye have led to a sorption capacity of 104.67 mg/g at 20 °C [70]. *Trichoderma viride* immobilized in calcium alginate was used in a continuous packed-bed column for the biosorption of Cr(VI), Ni(II), and Zn(II) ions from simulated aqueous solutions and electroplating effluent, obtaining a recovery efficiency of 40.1% for Cr(VI), 75% for Ni(II), and 53% for Zn(II) in five cycles of adsorption/desorption, observing that the process is enhanced by increasing the bed height in the column, decreasing the flow rate and the initial concentration of cations [61]. *Trichoderma asperellum* immobilized in calcium alginate was analyzed for Cu(II) biosorption, both in active and inactive form, obtaining a better biosorption efficiency when using inactive immobilized cells (134.22 mg/g) compared to active immobilized cells (105.96 mg/g) and polymer support (94.04 mg/g) as biosorbent at optimum pH between pH 4 and 5, and a desorption around 90% Cu(II) with HCl [62]. *Trichoderma harzianium* immobilized in Ca-alginate (stable for more than 8 weeks at experimental conditions) was used in a discontinuous system and a continuous adsorption column for removal and recovery of uranium ions from aqueous solutions. The results obtained in shake flasks proved an increased performance of the immobilized biomass compared to free biomass: 97.3%, 89.8%, and 87.5% in comparison with 75.3%, 56.0%, and 40.3% (temperature 28 ± 2 °C and agitation 200 rpm). For the system with only alginates as biosorbent, an efficiency of 42–43% was recorded, proving that through immobilization of biomass, the bioprocess efficiency is improved [71]. *Penicillium citrinum* from copper polluted sites was analyzed as biosorbent for Cu(II) from simulated synthetic wastewaters, obtaining a maximum sorption capacity of 25 mg/g compared to 22.4 mg/g for free biomass (biosorption efficiency 74.2% to 82.9% for immobilized biomass compared to 72.2% to 77.1% for free biomass), for the following conditions: pH 5.0, contact time of 20 min, and biosorbent dose of 1.5 g/L [72].

***Chitosan*** (β-(1→4)-2-amino-2-deoxy-d-glucose) is a hydrophilic, linear, semi-crystalline polymer obtained by chemical transformation (partial deacetylation) of chitin (a polysaccharide containing mostly β-(1→4)-2-acetoamido-2-deoxy-d-glucose that can be separated from shrimp, crab shell, other crustaceans, but also fungi). Chitosan is low cost, non-toxic, easy to process, is not abundant in nature but can be easily obtained from chitin, and possesses increased biosorption capacities compared to chitin, due to the presence of supplementary amino groups. Its particularities (natural positively charged hydrophilic polymer that can easily retain cells or proteins and can be physically/chemically modified) made it an appropriate choice in many areas (biotechnology, biomedicine, food industry) and allowed it to be processed in different ways: nanoparticles, flakes, gel beads, membranes, sponge, composite honeycomb, fibers, or hollow fibers [73,74]. Different microorganisms have been immobilized in chitosan and used for increasing its biosorption capacities: immobilized *Saccharomyces cerevisiae* in magnetic chitosan microspheres for the removal of Sr(II) from aqueous solution, with maximal adsorption capacity (q_m_) of 81.96 mg/g by the Langmuir model [74]. *Saccharomyces cerevisiae* immobilized on the surface of chitosan-coated magnetic nanoparticles was analyzed for the biosorption of Cu(II) from aqueous solutions, obtaining a maximal adsorption capacity of 144.9 mg/g following the same Langmuir model [75]. *Chlorella vulgaris* immobilized in mats of electrospun chitosan nanofibers was used in final effluents (especially wastewaters) treatments for inorganic phosphate and nitrate (initial 30 mg/L N-NO_3_^−^) removal, obtaining 87% N-NO_3_^−^ removal for the immobilized cells compared to 32% for the polymer support (chitosan) and a removal rate 12-fold higher compared to the free cells (non-immobilized) [76]. Vasilieva used cross-linked chitosan-based polymers (250 kDa or 600 kD cross-linked with glutaraldehyde) for *Lobosphaera* sp. IPPAS 2047 microalgae immobilization, proving a significant increase in nitrate (N_i_) and phosphate (P_i_) uptake efficiency for the immobilized cells (N*_i_*: 0.50 mg/mg Chl/d and P_i_: 0.36 mg/mg Chl/d for suspended cells compared to 0.65 mg Ni/mg Chl/d and 6.01 mg/mg Chl/d for immobilized cells) [77]. Analyzing the efficiency of total nitrogen removal from wastewaters for *Anabaena doliolum* and *Chlorella vulgaris* immobilized in chitosan flakes, alginate, agar, and carrageenan, a higher removal was observed for using chitosan as polymeric support for immobilization [78]. 

The use of a certain biopolymer can change the morphology of the final biosorbent particles. For microencapsulation by spray-drying, for example, in Figure 3, the microparticles surface has different textural characteristics, but the same diameter (around 3 μm); for chitosan used as biopolymer, the surface was very rough with a characteristic surface structure with a concavity, while for sodium alginate the surface was smooth and for chitosan the microparticles presented a very regular shape and a smooth surface.

In Table 4 are presented some examples of microbial biomasses that have been used for specific removal of contaminants. From these data it can be observed that alginate-based biosorbents are mostly used for heavy metals removal and for anionic dyes, while chitosan-based biosorbents were mostly used for cationic dyes removal.

## 4. Microbial Biomass-Based Biosorbents

Microbial biomass-based adsorbents can be used for the removal of pollutants as metals, dyes, or antibiotics, and are separated mainly under the following categories: algae, bacteria, or fungi that can be immobilized on different polymeric supports [6,101,102]. Their structure, properties, and chemical composition strongly influence the biosorption efficiency. Table 5 presents some general aspects about different types of biosorbents, regarding the preparation/synthesis.

### 4.1. Algal Biomass

The biosorption efficiency depends, in the case of algal biomass, on the cell wall components (chitin, glycan, cellulose, and alginate, which can bind and remove the pollutants), cell surface, and spatial structure [54]. For metal removal the algal cells use the following functional groups found in natural algae form that acts like active sites for metal binding: the carboxyl group (the most important), sulphate, amino, hydroxyl, carbonyl, thioester, phosphodiester, amine, and amide. By applying different pretreatments (chemical—acid/alkali, salts, organic compounds that can dissolve the outer cell surface) the stability, but also the number of functional groups of the algal biomass, can be increased and biosorption efficiency can be improved. The algae immobilization for effluent (wastewater) treatment provides many advantages related to improvement of the hydrodynamic behavior and the physical strength of the micro-particles (due to the cellulose from the algae cell-wall), but some disadvantages must be taking into account: mass transfer limitation through supplementary diffusion and inactivation of some biomass active groups due to support binding or other cells [103,104]. *Spirulina* immobilized on alginate beads was used as biosorbent for lead removal in batch and fixed-bed columns, higher removal capacity being observed at pH 5.2—adsorption capacity of 114.47 mg Pb(II)/g [105]. *Chlorella* sp. and *Chlamydomonas* sp. immobilized in the form of sodium alginate beads showed improved biosorption abilities for copper ions compared to free biomass: 33.4 and 28.5 mg/g for *Chlorella* sp., while for zinc ionic species the results were similar for free and immobilized biomass and lower than for copper ions (around 26 mg/g), due to different functional groups that bind the cations in the two strains [106].

### 4.2. Bacterial Biomass

Bacteria are unicellular microorganisms, available in different shapes (cocci, rods, spiral, and filamentous) that contain cell wall, cell membrane, cytoplasm, and an ADN (deoxyribonucleic acid) chain, that are divided as gram-positive and gram-negative due to differences in the cell wall. The main component of bacterial cell wall is peptidoglycan, a polymer containing N-acetylglucosamine and N-acetylmuramic acid that provides the cell form and rigidity (Gram-positive bacteria possess a thick peptidoglycan layer—90% of the cell wall, while Gram-negative bacteria possess a thin peptidoglycan layer—20%). At this level, several functional groups (observed through Fourier transform infrared spectroscopy) are available for biosorption: carboxyl, phosphoryl, hydroxyl (involved in the sorption of metals), or amine (involved in the sorption of organic compounds (dyes, antibiotics) through electrostatic interaction) that can attach the pollutants [80]. The anionic functional groups are present in the peptidoglycan, teichoic acids, and teichuronic acids in the case of Gram-positive bacteria, and the peptidoglycan, phospholipids, and lipopolysaccharides in the case of Gram-negative bacteria and can bind especially metal cations [107].

### 4.3. Fungal Biomass

Fungi include diverse groups of unicellular (yeast) and multicellular (molds) microorganisms, that contain cell wall, cell membrane, cytoplasm, a nucleus, and several cellular organelles (ribosomes, endoplasmic reticulum, Golgi apparatus, mitochondria, etc.). The cell wall, consisting mainly of polysaccharides (glucan—28%, mannan—31%, chitin and chitosan—2%), but also proteins—13%, lipids—8%, is rigid and is responsible for the cell’s structure and shape. In the biosorption process, this cellular feature offers the functional groups for the pollutant binding (amide, amine, carbonyl, carboxyl, hydroxyl, imine, imidazole, sulfonate, sulfhydryl, thioether, phenolic, phosphate, and phosphodiester groups) [108]. For fungal cell immobilization, entrapment and crosslinking are the most used, as they provide several advantages: biomass can be loaded in large volume, appropriate/adjustable particle size, easy separation and minimized clogging, and high regeneration [109]. Fungal biomass is easily available as industrial byproducts from biotechnological processes used for the production of antibiotics (*Penicillium* sp.), organic acids (*Aspergillus* sp., *Rhizophus* sp.) and from the brewery industry (*Saccharomyces cerevisiae* (yeast)).

### 4.4. Factors Influencing the Batch Biosorption

Adsorption is a spontaneous process where different interactions occur between adsorbent and adsorbate, and it can be divided in four types: ion exchange (attachment to the adsorbent surface of ionic species of opposite charge), physisorption (interaction occurs through weak van der Waals forces through the adsorbent surface and the non-ionized adsorbate), chemisorption (involves strong chemical bonds between adsorbent surface and dissociated adsorbate that implies a change in the chemical form of adsorbate), and specific adsorption (occurs through specific interaction between adsorbate and adsorbent without any chemical change of adsorbate) [6,7,8,19]. In Figure 4 the schematic representation of biosorption mechanism for heavy metals and dyes is presented. For metal biosorption the most dominant mechanism is ion exchangeand and also compplexation/chelatation, a process in which the biosorbent functional groups that can bind metals are carboxyl, sulfate, hydroxyl, and phosphoryl. For cationic metals a pH between 7 and 8 is recommended in order to avoid hydrogen competition for biosorbent’s active site at lower pH, while for anionic metals (chromium, molybdenum) the recommended pH is between 2 and 4, which generates more positive ionized groups that would bind the anions. For dyes removal electrostatic interaction (beside hydrogen bonding, electrostatic, hydrophobic, and van der Waals interaction) plays an important part: for cationic dyes (Crystal violet, Methylene Blue, Rhodamine B) the interaction is established between the negative charge of the biosorbent surface and the positive charge of the cationic dye, while for anionic dyes (Reactive Blue 19, Eriochrome Black T) the interaction is between the positive charged biosorbent and the anionic dye.

The main four factors that affect biosorption are: physico-chemical factors such as solution pH (it determines the ionization of both biosorbent and pollutant), ionic strength (with ionic strength increase, biosorption efficiency decreases due to competition with the adsorbate for biosorbent’s binding sites), temperature (by increasing temperature, the activity and kinetic energy of the adsorbate increases, improving the biosorption efficiency, but high values of temperature can affect the physical structure of the biosorbent), agitation rate (the agitation reduces the diffusion step resistance in the mass transfer, improving the biosorption efficiency, but high values can affect the physical structure of the biosorbent), biomass dosage (larger amounts of biomass will offer an increased biosorption area which improves the process efficiency, but usually higher yields are obtained at low biosorbent quantities, as there is less interference between binding sites), initial solute concentration (an increased solute concentration usually provides a low biosorption efficiency, but an increased pollutant quantity retained per unit of biosorbent), and nature and type of biomass (different cells offer different functional groups for the biosorption and different contact areas).

A key parameter that strongly influences the large scale use of biosorbents is their reusability, which implies effective regeneration through desorption (use of eluents as alcohols, acids, or alkali) that would make it possible to reuse them for a large number of cycles [11]. The selection of an appropriate eluent is of critical importance since it needs to fulfil several requirements: avoid affecting the biosorbent’s structure, increase affinity but also easy separation from the adsorbate, and be eco-friendly and inexpensive. For dyes, ethanol and methanol are used as eluents while for heavy metals the most commonly used eluent is hydrochloric acid, but also sulfuric acid, sodium hydroxide, and also EDTA could be used for desorption [54]. Experimental results showed that through regeneration some loss in the biosorption capacity is recorded: for gold ionic species 60% after three biosorption cycles using *Lysinibacillus sphaericus* encapsulated into alginate matrix [110], for Eu(III) biosorption using *Saccharomyces cerevisiae* immobilized on the chitosan matrix the biosorbent was used for four cycles before major loss of biosorption capacity [111].

The biosorption efficiency and selectivity is strongly affected by the presence of other ions or organic pollutants due to competitive adsorption [112]. In Ag(I) and Cu(II) batch system biosorption an increased equilibrium time of 10 times (from 60 to 600 minutes) was recorded at higher copper initial concentration, but the equilibrium time for Cu (II) (75 min) was not affected by the silver initial concentration in kinetic experiments using brown algae waste [113]. The study of competitive biosorption of Yellow 2G and Reactive Brilliant Red K-2G using inactive aerobic granules in binary solutions resulted in maximum biosorption capacity of 58.50 and 66.18 mg/g respectively in single solution, while in binary solutions, the biosorption efficiency decreased to 40.38 mg/g for Yellow 2G, and increased to 171.21 mg/g, for Reactive Brilliant Red K-2G, due to a smaller molecular size and shape that enhanced the dye penetration to the internal biosorbent structure [114].

Despite the importance of the matter, there are few studies in the literature addressing pollutants biosorption applied for real wastewaters using immobilized microbial biomass, but some recent data is available for free biomass. The brown algae *Sargassum filipendula* was used for Al removal from different effluents that contained in different concentration: aluminum, chromium, lead and zinc: wastewater from a tannery industry, wastewater from a treatment facility of leather industry, the entrance of a French urban water treatment station in Strasbourg and the exit of the same water treatment. The results showed that the biosorbent was able to retain Al ionic species 43%, 81%, and 44% respectively for the first three water sources, while for the last one no removal was noted due to a low Al ionic concentration (<0.02 mg/L) [115].

## 5. Conclusions and Future Perspectives

The biosorption is a technology that offers a great diversity of options and combinations for the elimination of pollutants, demonstrating great flexibility for its application for the removal of metals, dyes, antibiotics from effluents (wastewaters), but with limited practical application due to complexity of operation. This is due to the presence of multiple pollutants in the wastewaters that makes the process more difficult; however, the appropriate selection of both support polymer and immobilized microbial biomass could increase the biosorption practical application. The possibility of using wasted biomass (by-products from different industries), in free or immobilized form, offers advantages in terms of cost but also in durability/sustainability and environmental protection. The microorganisms act as biosorbents binding the pollutants, while the support polymer protects the microbial biomass, but can also provide functional groups for the biosorption, thereby improving the process. Analyzing the reviewed biosorbents, alginate (most commonly employed polymer in the biosorption process due to its non-toxicity, increased stability, and easy generation) and chitosan-based biosorbents showed the best biosorption capacities for pollutant removal. Taking this into account, research should be focused on the development of better carriers for microorganism’s immobilization: cost-effective and biocompatible, stable from a physico-chemically point of view, providing sufficient adsorption sites. Also of great importance is the selection of appropriate microorganisms and more knowledge on the interactions between the support polymers and the biomass, for increase stability and lower operation costs. The polymers chosen for the immobilization of the microbial biomass need constant attention, as their properties strongly influence the biosorption efficiency, the form of the biosorbent, and its properties. Natural polymers are preferred as polymeric support as they are widely available, but their properties need modifications. Thus, derivate polymers are newly used for different immobilization techniques, due to tailored properties. Even if biosorption offers many advantages, there are several challenges related to the development of large-scale procedures of biosorbents synthesis, but combining biomass and polymeric materials can lead to obtaining biosorbents that would improve the process effectiveness and increase its application in real conditions. Research should be focused on experiments using real wastewaters on large scale (influence of process parameters in different industrial systems) necessary to transform biosorption into a large scale competitive, commercially efficient process. Additionally, the newly investigated biosorbents need to be evaluated not only in regards to their biosorption efficiency, but also their physical straight, regeneration capacity, and desorption efficiency, as for biosorbent (especially obtained through immobilization) reusability is essential. It is also important to compare them to commercially available sorbents in order to establish a hierarchy used for choosing the best biosorbent for a specific purpose. Microorganisms have the ability to adjust to different conditions, sometimes extremely hard ones, and scientists possess the knowledge to transform them in order to increase their abilities to transform or to adsorb contaminants; however, these new strains need testing in order not to introduce in the environment potentially dangerous strains (especially in the case of living microorganism). The cost of producing eco-friendly biosorbents is also important, as besides high efficiency and availability combined with promising regeneration, low cost is extremely important. There is also a lack of studies in the literature regarding the use of biosorption technology in continuous systems (e.g., packed-bed column), even if the continuous mode offers more advantages compared to the batch one, especially for the expansion at industrial scale.

## Data Availability

The study did not report any data.

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
