# Peer review of "Polysaccharides as Support for Microbial Biomass-Based Adsorbents with Applications in Removal of Heavy Metals and Dyes"

_polymers, 2021, doi:10.3390/polym13172893_

Round 1

Reviewer 1 Report

The document is a revision on the use of biosorbents to be used in the removal of contaminants in industrial and domestic wastewaters.  The subject is of interest, and is well covered by the authors. 

They focus on two areas: the use of biomass from microorganisms as biosorbents, and the use of polysaccharides as entrapment material. They include diverse examples of the latter. 

I have only a few suggestions to  improve de document before publication: 

Plant biomass as biosorbent material has the disadvantage of lining content, and the need for a chemical process in order to remove it. This adds to the environmental and economic cost on the use of this residual product. This is not mentioned in the document, and I think is important to describe it. 

The other is a suggestion to better present the data. Would it be possible to add a table with examples of microbial biomass (bacteria, fungi) that has been used for specific removal of contaminants, selected from the information incorporated in the document. 

Author Response

Please see the attachmnent

Reviewer 2 Report

This article reviewed the applications of polymers assisted biosorbents for applications in water treatment. Unfortunately, the manuscript does not contain sufficient data to support the title, nor detailed and critical review is provided. I cannot recommend it for publication in this form.

Below are some specific comments. 

  1. The abstract is very general. The novelty of work, methodology, the need of this review, and key findings/recommendations/gaps are not provided.
  2. The manuscript is mainly focused on biosorption. I don’t know where the authors have incorporated the polymers based biosorbents in the text?
  3. Section 3 is somehow relevant to the topic. However, it is not organized. What are the various techniques for incorporating polymers into biosorbents? What parameters affect the synthesis process? What are gaps? Which method is best? What are the authors recommendations? Schematics of synthesis methods should also be provided.
  4. What are the specific characteristics of biosorbents improved after polymer addition? A detailed comparative analysis with support from the literature should be provided.
  5. What types of pollutants are removed by polymer-based biosorbents? A comprehensive list of tables should be provided either based on pollutant type, biosorbent or polymer. The experimental conditions and biosorbent performance (capacity), mechanism of adsorption, regeneration potential should be portrayed in detail.
  6. Schematics of mechanisms should be provided.
  7. Future recommendations are a critical part of the review. It should be provided.
  8. Being a review paper, more references need to be provided but make sure it should be recent and relevant.

Reviewer 3 Report

The review is interesting. However, the authors can further modify the draft . Folllowing suggestions are given.

The title does not reflect the content of the review. Polymers should be replaced by polyscahharides or other suitable word that reflect the content of the review.

The authors should summarize in the form of a table which biopolymer is sued for which type of metal removal. Foe example the polymers used for removal of Cu and so on.

Round 2

Reviewer 2 Report

The revised version of the manuscript is somehow improved. However, still, a significant modification is needed before co considering it for further processing.

  1. The abstract is still very general with mainly discussing the benefits of bio-sorbents. It should also mention how this review is organized? What are the research gaps? What are the authors recommendations?
  2. The introduction is very brief. It should be extended with more literature coverage and should include the latest literature on the topic
  3. Scheme 1 can be modified with a clear description of pollutants and bio-sorbent surfaces.
  4. Use a uniform terminology through the text, such as Ni2+ or Ni(II).
  5. A separate section on the research gap and future recommendations can be included.
  6. The tile is very general, i.e., the removal of pollutants. I will suggest keeping it a bit specific, i.e. focusing on heavy metals or dyes or both only.
  7. The interaction mechanism of specific pollutants should be discussed in a separate section with some good schematics from published literature (after taking copyright permission).
  8. How the adsorbents are regenerated and used in multiple runs. Relevant literature should be included.
  9. How the process efficiency is affected by the presence of other pollutants. Specific examples from published data.
  10. Are these bio-sorbents used in real water treatment? Or for simultaneous removal of multi-pollutants? Discuss with examples from literature.
